

# Cells released from *S. epidermidis* biofilms present increased antibiotic tolerance to multiple antibiotics

Vânia Gaio and Nuno Cerca

Laboratory of Research in Biofilms Rosário Oliveira—Centre of Biological Engineering, University of Minho, Braga, Portugal

## ABSTRACT

Biofilm released cells (Brc) are thought to present an intermediary phenotype between biofilm and planktonic cells and this has the potential of affecting their antimicrobial tolerance.

**Aim**. Compare the antimicrobial tolerance profiles of Brc, planktonic or biofilm cultures of *S. epidermidis*.

**Methodology**. Planktonic, biofilm cultures or Brc from 11 isolates were exposed to peak serum concentrations (PSC) of antibiotics. The antimicrobial killing effect in the three populations was determined by CFU.

**Results**. Increased Brc tolerance to vancomycin, teicoplanin, rifampicin, erythromycin, and tetracycline was confirmed in model strain 9142. Furthermore, significant differences in the susceptibility of Brc to vancomycin were further found in 10 other clinical isolates.

**Conclusions**. Brc from distinct clinical isolates presented a decreased susceptibility to most antibiotics tested and maintained that enhanced tolerance despite growing planktonically for up to 6 h. Our data suggest that Brc maintain the typical enhanced antibiotic tolerance of biofilm populations, further suggesting that addressing antimicrobial susceptibility in planktonic cultures might not reflect the full potential of biofilm-associated bacteria to survive therapy.

## INTRODUCTION

The interest in studying *Staphylococcus epidermidis*' pathogenicity raised over the last decades, especially since this microorganism has been associated with an increasing number of infections allied with the use of indwelling medical devices (*Heilmann, Ziebuhr & Becker, 2018*). *S. epidermidis* was primarily seen as a commensal microorganism owing to its non-infectious lifestyle, benign relationship with the host (*Schoenfelder et al., 2010*; *Gardiner et al., 2017*) and maintenance of a healthy skin microflora (*Cogen, Nizet & Gallo, 2008*). However, *S. epidermidis* is now acknowledged as an opportunistic pathogen, causing a wide range of infections, especially in ill and immunocompromised patients and neonates (*Otto, 2009*). Since *S. epidermidis* is ubiquitously present in human skin and mucosae, this species present an inherent facility to overcome some compromised physiological

Corresponding author
Nuno Cerca,
nunocerca@ceb.uminho.pt

barriers (*Ziebuhr et al., 2006*). Furthermore, *S. epidermidis* is well described as a thick and multi-layered biofilm forming species (*Cerca et al., 2005b*). Biofilms are commonly defined as an organized aggregation of microorganisms and their extracellular products, i.e., a well-structured population of microbes embedded in a self-produced matrix of polymeric substances, generally attached to a surface (*Costerton et al., 1987*). Due to the great impact of biofilms in infections, in particular its higher antimicrobial resistance (*Mah, 2012*; *Dias et al., 2018*), the process of biofilm formation has been extensively studied in the past decade. Nowadays, it is recognized as a controlled process that comprises multiple steps, commonly divided into three main phases: attachment, maturation and disassembly (*Otto, 2013*). During the biofilm lifecycle, a small number of cells is continuously released from the biofilm to the surrounding environment; moreover, major events of cell releasing may occur when the biofilm reaches a fully developed state, phenomenon acknowledged as biofilm disassembly (*Boles & Horswill, 2011*; *Mack et al., 2007*). This process is associated with the occurrence of serious infections (*Boles & Horswill, 2011*) and may be triggered by shear forces and abrasion of the biofilm structure (a process known as detachment) (*Choi & Morgenroth, 2003*). On the other hand, it can be prompted by environmental conditions, as nutrient and oxygen depletion (*Hunt et al., 2004*), pH and temperature changes (*Boles & Horswill, 2011*) and accumulation of waste (*Kaplan, 2010*) (a process known as dispersion). Biofilm released cells (Brc) have the ability to colonize different sites in the host upon release, contributing to the spreading of local infections (*Kaplan, 2010*), and occurrence of systemic diseases, as bacteraemia (*Cervera et al., 2009*). Interestingly, *S. epidermidis* has a great ability to adhere to endothelial cells, contributing to bloodstreams infections and to its pathogenicity (*Merkel & Scofield, 2001*). Bacteraemia is the main staphylococcal systemic infection and one of the major causes of morbidity and mortality among hospitalized patients with chronic diseases and/or compromised immune systems (*Kleinschmidt et al., 2015*; *Krcmery et al., 1996*). Important key features of *S. epidermidis* biofilms are their higher tolerance to antimicrobial therapies (*Cerca et al., 2005a*; *Cerca et al., 2005b*; *Albano et al., 2019*), and their ability to evade the host immune defences (*Cerca et al., 2006*). Since it was previously shown that the release of cells is related with the emergence of serious acute infections (*Boles & Horswill, 2011*), as Brc may enter the blood circulation and cause systemic diseases (*Cervera et al., 2009*) or spread around the body and cause local infections (*Otto, 2013*), it is of urgent nature to deeply study this population of cells. Noteworthy, recent evidence suggests that Brc from *S. epidermidis* strain 9142 present a distinct phenotype from both biofilm and planktonic cells, with implications in antimicrobial tolerance to vancomycin and tetracycline (*França et al., 2016a*) and also in the adaptation to the host immune system (*França et al., 2016b*). Therefore, there is also an urge to study the distinct antimicrobial tolerance profiles of the distinct populations of cells that may be found in biofilm-related infections, i.e., biofilm, Brc and, eventually, planktonic cells, as common antimicrobial therapies may not consider the increased tolerance to antibiotics of some populations and, thus, fail in treating the infection. To better assess the implications of Brc in enhanced tolerance to antibiotics, a battery of nine antibiotics was tested using model strain *S. epidermidis* 9142. Furthermore, in order to understand if this altered phenotype is widespread among *S. epidermidis* clinical isolates,

**Table 1** Description of the 11 *Staphylococcus epidermidis* isolates used in this study.

| *S. epidermidis* isolate | Description | Country of origin | Sequence typing / Clonal Complex (*Miragaia et al., 2007*; *Cerca et al., 2013*) |
|---|---|---|---|
| RP62A (ATCC 35984) (*Christensen et al., 1985*) | Clinical isolate from catheter-associated sepsis | United States of America | 10 / 2 |
| 9142 (*Mack, Siemssen & Laufs, 1992*) | Clinical isolate from blood culture | Germany | 10 / 2 |
| IE186 (*Cerca et al., 2004*) | Clinical isolate from a patient with infective endocarditis | United States of America | 367 / 2 |
| PT12003 (*Freitas et al., 2017*) | Clinical isolate from a patient with gastric disease | Portugal | Unknown |
| 1457 (*Mack et al., 1994*) | Clinical isolate from a venous catheter-associated infection | United States of America | 86 / 2 |
| DEN69 (*Cerca et al., 2013*) | Unknown | Denmark | 56 / S56 |
| URU23 (*Cerca et al., 2013*) | Unknown | Uruguay | 86 / 2 |
| IE214 (*Cerca et al., 2006*) | Clinical isolate from a patient with infective endocarditis | United States of America | 10 / 2 |
| PT13032 (*Freitas et al., 2017*) | Clinical isolate from a patient with chronic renal failure | Portugal | Unknown |
| ICE09 (*Cerca et al., 2013*) | Unknown | Iceland | 6 / 2 |
| MEX60 (*Cerca et al., 2013*) | Unknown | Mexico | 61 / 2 |

Brc tolerance was also assessed using clinical isolates from distinct parts of the world, when exposed to vancomycin.

# MATERIALS & METHODS

## Antibiotics and bacterial isolates

Antibiotics with three distinct mechanisms of action (cell wall, nucleic acids and protein synthesis inhibitors) were selected to conduct this analysis and were used at their peak serum concentrations (PSC) (Table S1). A total of 11 *S. epidermidis* isolates, previously characterized, were used in this study (Table 1).

## Growth conditions

A previously described model (*França et al., 2016a*; *França et al., 2016b*) was followed in this study. Briefly, an inoculum was done by adding one *S. epidermidis* colony into 2 mL of Tryptic Soy Broth (TSB) (Liofilchem, Teramo, Italy) and incubated in an orbital shaker overnight at 37 °C and with agitation at 120 rpm. Later, the overnight cells were diluted in TSB medium until an optical density (OD) at 640 nm of $0.250 \pm 0.05$ was reached, corresponding to an approximate concentration of $2 \times 10^8$ CFU/mL (*Freitas et al., 2014*). Biofilms were formed through the inoculation of 15 $\mu$L of the adjusted suspension into a 24-well microtiter plate (Orange Scientific, Braine-l'Alleud, Belgium), with 1 mL of TSB supplemented with 0.4% (v/v) glucose (TSBG) to induce biofilm formation, being incubated at 37 °C with shaking at 120 rpm, for as long as 48 ($\pm$1) hours in an orbital shaker. After 24 ($\pm$1) hours of incubation, spent medium was carefully removed and

 

biofilms were washed twice with a saline solution (0.9% (m/v) NaCl in distilled water) to remove unattached cells, followed by careful addition of 1 mL of fresh TSBG and subsequent incubation in the same conditions. Finally, at each time point, the supernatant was removed, biofilms were washed twice with a saline solution and suspended in 1 mL of the same by scraping cells from the plastic surface. Disrupted biofilm cells were pooled together from at least four distinct wells to decrease biofilm formation variability (*Sousa, França & Cerca, 2014*). Planktonic cells were grown for 24 (±1) hours at 37 °C and with 120 rpm agitation in an orbital incubator. It was previously shown (*França et al., 2016a*) that, after discarding the spent media, most of the non-adherent biofilm cells were washed away with the NaCl washing procedure done twice. Thus, cells grown on the suspension after the media replacement are mostly cells released from the biofilm (Brc), that were collected by careful aspiration of the biofilm bulk fluid 24 h after media replacement performed on 24-h old biofilms.

## Homogenization and quantification of bacterial populations

Prior to any assay, the three populations (disrupted biofilm cells, Brc and planktonic cells) were submitted to a pulse of 5 s of sonication with 40% amplitude (Ultrasonic Processor Model CP-750, Cole-Parmer, Illinois, USA) to homogenize the suspensions and disassociate possible existing clusters. This sonication cycle did not have a significant effect on cell viability while being able to dissociate the majority of the clusters and significantly reducing the size of the remaining ones, as previously demonstrated (23). The bacterial populations were quantified by OD measurement at 640 nm ($OD_{640}$) (*Freitas et al., 2014*). At least three independent experiments, with technical duplicates, were performed.

## Comparison of the antibiotic susceptibility of the distinct *S. epidermidis* populations assessed by CFU counting

Bacterial populations were diluted in TSB reaching a final concentration of about $2 \times 10^7$ CFU/mL. Then, each antibiotic was added to the previous suspension at the respective PSC and the assay tubes were incubated at 37 °C with agitation at 120 rpm for up to 6 h. Simultaneously, controls were performed by inoculating the same suspensions in TSB, without adding antibiotics. All the tubes were prepared in duplicate. After 2 and 6 h of incubation, 1 mL of each tube was collected and centrifuged at 4 °C and 16,000 g for 10 min. Next, the supernatant was carefully discarded and the pellet was suspended in 1 mL of saline solution, with the aid of a pulse of 5 s of sonication at 40% amplitude. Finally, 10-fold serial dilutions were performed and plated onto Tryptic Soy Agar (TSA) plates to allow CFU counting. The experiments were performed at least three independent times, with technical duplicates.

## Determination of the minimum inhibitory concentration (MIC)

The MICs were determined according to NCCLS standards (*National Committee for Clinical Laboratory Standards, 1997*), with some minor modifications, using planktonic cells and TSB as a growth medium and performing at least two consistent independent assays, with technical duplicates for each determination. Moreover, the results obtained in the MIC assays were then compared with the clinical breakpoints for *S. epidermidis* described in the

literature. The European Committee on Antimicrobial Susceptibility Testing (EUCAST) (*European Committee on Antimicrobial Susceptibility Testing, 2016*) was followed for the majority of the antibiotics, and the Clinical and Laboratory Standards Institute (CLSI) (*Clinical and Laboratory Standards Institute, 2015*) was followed for dicloxacillin since EUCAST did not provide the MIC breakpoints for this antibiotic.

### Statistical analysis

Statistical significance between control and antibiotic-treated samples was determined using one-way ANOVA multiple comparisons ($p < 0.05$). All analysis was performed using GraphPad Prism version 6 (Trial version; San Diego, CA, USA).

## RESULTS

### Profile of Brc, biofilm and planktonic cells antimicrobial tolerance to multiple antibiotics

To deepen the knowledge of *S. epidermidis* Brc physiology, in particular their antimicrobial tolerance, and improve the relevance of our previous findings, the number of antimicrobials and isolates was broadened in this study. First, nine antibiotics, representing three distinct mechanisms of action, were selected to be tested against model strain *S. epidermidis* 9142. The PSC for each antibiotic was used, since this concentration refers to the maximum amount of antibiotic the human body can endure after common antimicrobial therapy, having an important clinical relevance for the comparison of the antimicrobial effect among distinct antibiotics. The antimicrobial activity was assessed against Brc, biofilm disrupted cells and stationary phase planktonic cells. As shown in Fig. 1, significantly increased tolerance of Brc to antibiotics, as compared to stationary planktonic cells, was found with vancomycin, teicoplanin, rifampicin, erythromycin and tetracycline. This effect was antibiotic specific since it was not observed with dicloxacillin, gentamicin and linezolid. Furthermore, ciprofloxacin had no effect in either biofilms or Brc, and, as such, could not be used in this comparison. The lower efficiency of ciprofloxacin and linezolid could be explained by the higher MIC values (Table S2).

### Comparative tolerance to vancomycin by the 3 distinct bacterial populations in multiple *S. epidermidis* isolates

In earlier reports (*França et al., 2016a*; *França et al., 2016b*), Brc were found to present a significant increased antimicrobial tolerance to vancomycin. As there are still few isolates resistant to vancomycin (*Pinheiro et al., 2015*), we chose this antibiotic for testing Brc tolerance in multiple clinical isolates, representative of different regions of the world (*Cerca et al., 2013*; *Miragaia et al., 2009*). We first confirmed that all isolates were susceptible to vancomycin (Table S3). Next, the susceptibility of Brc, stationary phase planktonic cells and biofilm-disrupted cells to PSC of vancomycin was evaluated. As represented in Fig. 2, the phenotype of increased vancomycin tolerance found with Brc was confirmed in 10 out of 11 isolates, confirming that this phenomenon occurs in distinct *S. epidermidis* isolates, most from the clonal complex 2, which is the main clonal lineage in hospitals worldwide (*Miragaia et al., 2007*; *Du et al., 2013*).

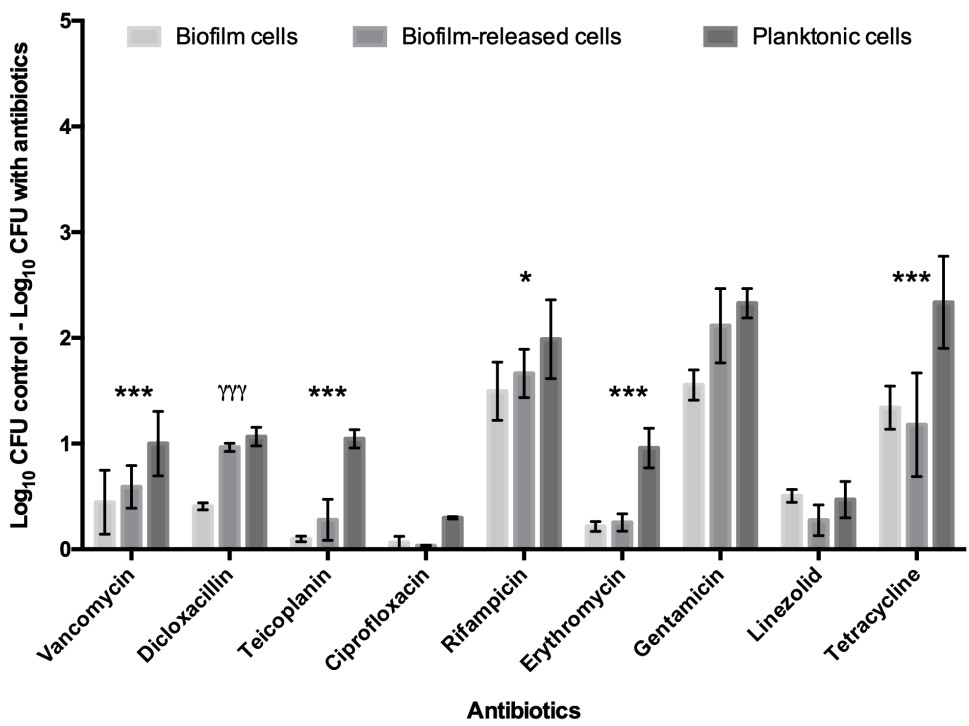

**Figure 1** **Base 10 logarithmic CFU/mL reduction of *S. epidermidis* 9,142 populations upon 2 hours of incubation with PSC of distinct antibiotics.** The columns represent the mean plus or minus standard error deviation, of at least three independent experiments. Statistical differences between groups were analysed with one-way ANOVA multiple comparisons Statistically significant differences between biofilm cells and Brc are represented with * (* $p < 0.05$; *** $p < 0.001$), and between Brc and their planktonic counterparts with $\gamma$ ($\gamma \, \gamma \, \gamma \, p < 0.001$).

## DISCUSSION

When biofilms release cells to the bloodstream, the chance for developing acute infections arises (*Boles & Horswill, 2011*). Despite the relevance of biofilm disassembly, there are very few studies addressing how these cells interact with the host and with antimicrobial therapy. In a recent study, it was shown that Brc from *S. epidermidis* strain 9142 had distinct phenotypic features, including increased tolerance to vancomycin and tetracycline (*França et al., 2016a*; *França et al., 2016b*). Herein, we aimed to deepen those previous findings by assessing if the previous observable phenomenon was antibiotic specific or strain specific.

By using a battery of nine antibiotics from different classes, we concluded that Brc present an increased tolerance to several antibiotics that are commonly used to treat *S. epidermidis* infections. Interestingly this phenomenon is antibiotic specific since not all tested antibiotics had a lower efficiency against Brc. Thereby, our data suggest that antibiotics in which a similar tolerance was found among Brc and planktonic cells, such as dicloxacillin, gentamicin and linezolid, may present a better therapeutic alternative when *S. epidermidis* biofilm infections are present. Although no differences were observed in the susceptibility of Brc and planktonic cells to ciprofloxacin, the results from this antibiotic can be considered an exception, since strain 9142 was naturally resistant to this antibiotic.

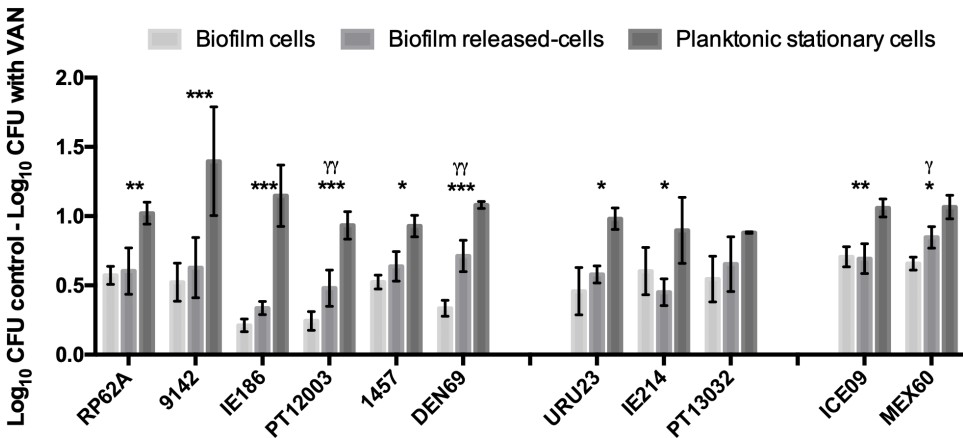

**Figure 2** **Base 10 logarithmic CFU/mL reduction of the three populations of *S. epidermidis* isolates after 2 h of incubation with PSC of vancomycin.** The columns represent the mean plus or minus standard error deviation, of at least three independent experiments. Statistical differences between groups were analysed with one-way ANOVA multiple comparisons. Statistically significant differences between biofilm cells and Brc are represented with * (* $p < 0.05$; *** $p < 0.001$), and between Brc and their planktonic counterparts with $\gamma$ ($\gamma$ $p < 0.001$).

Importantly, our data confirm that biofilm enhanced tolerance does not depends only on biofilm structure (*Singh et al., 2010*; *Zheng & Stewart, 2002*; *Jefferson, Goldmann & Pier, 2005*), since free-floating bacteria originated from a biofilm maintained enhanced tolerance to different antibiotics for several hours (Fig. S1).

Moreover, our data confirmed that altered vancomycin enhanced tolerance by Brc was common among the tested clinical isolates. Since vancomycin is an antibiotic which mechanism of action is related to targeting cell wall, this antimicrobial agent is more effective against actively growing cells (*Falcón et al., 2016*; *Kohanski, Dwyer & Collins, 2010*). It is probable that Brc enhanced tolerance to vancomycin may be partially related to growth rate since it was shown before that Brc can grow at higher rates than biofilm cells, upon adaptation to the suspension mode, but still at a lower rate than planktonic cells (*França et al., 2016a*; *França et al., 2016b*). However, differences in the duplication rates might only partially explain increased Brc tolerance to antibiotics, since the same effect was found in using rifampicin in *E. coli*, which antimicrobial effect does not depend on the duplication time (*Skarstad, Boye & Steen, 1986*).

## CONCLUSIONS

Overall, our results are in accordance with previous findings for distinct bacterial species, where the idea of Brc regaining susceptibility to antibiotics as soon as they are in planktonic form after being released from the originating biofilm, is questioned (*Boles & Horswill, 2011*; *França et al., 2016a*; *França et al., 2016b*; *Mack et al., 1994*; *Chua et al., 2014*). Moreover, this study broadens the spectrum of antibiotics to which *S. epidermidis* Brc were found to present an increased tolerance, as well as the finding of a wider range of clinical isolates with increased tolerance of Brc to vancomycin. How this data might reflect in vivo situations

is, yet, unexplored. It does provide insights regarding the occurrence of biofilm-related bacteraemia and how Brc are better adapted against antimicrobial therapy than what in vitro planktonic assays might point out. Perhaps, future antimicrobial susceptibly testing against *S. epidermidis* biofilm-related infections should be revised, in order to incorporate Brc suspensions instead of faster growing planktonic cultures. Future studies will help highlight how *S. epidermidis* specific phenotypes contribute to higher rates of tolerance towards common antimicrobial therapies.

### Funding

This study was supported by the Portuguese Foundation for Science and Technology (FCT) under the scope of the strategic funding of UID/BIO/04469/2019 and by BioTecNorte operation (NORTE-01-0145-FEDER-000004) funded by European Regional Development Fund under the scope of Norte2020 - Programa Operacional Regional do Norte. The funders had no role in study design, data collection and analysis, decision to publish, or preparation of the manuscript.

### Grant Disclosures

The following grant information was disclosed by the authors:
Portuguese Foundation for Science and Technology (FCT): UID/BIO/04469/2019.
BioTecNorte operation: NORTE-01-0145-FEDER-000004.
European Regional Development Fund under the scope of Norte2020 - Programa Operacional Regional do Norte.

### Competing Interests

The authors declare there are no competing interests.

### Author Contributions

- Vânia Gaio performed the experiments, analyzed the data, prepared figures and/or tables, authored or reviewed drafts of the paper, approved the final draft.
- Nuno Cerca conceived and designed the experiments, analyzed the data, authored or reviewed drafts of the paper, approved the final draft.

### Data Availability

The raw data of all figures and supplementary figures are available in the Supplemental Materials.

### Supplemental Information

Supplemental information for this article can be found online at http://dx.doi.org/10.7717/peerj.6884#supplemental-information.

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
