# Peer review of "Cells released from S. epidermidis biofilms present increased antibiotic tolerance to multiple antibiotics"

_PeerJ, doi:10.7717/peerj.6884_

## Round 0.1 · original submission · Minor Revisions

Both reviewers find that it is necessary to provide a few more details about how biofilm resealed cells (BRC) were obtained. Other comments also add to improve the formatting and clarity of the text.

·

Basic reporting

The manuscript entitled "Cells released from S. epidermidis biofilms present
increased antibiotic tolerance to multiple antibiotics" (#34906) compared the antimicrobial tolerance profiles of Brc, planktonic or biofilm cultures of S. epidermidis to different antibiotics. The data is consistent and it will provide good support for further works in the future. Moreover, the text is clear, well structured and provides sufficient basement for the purpose. Therefore, I support the article publishing with minor revisions as followed:

Brc is used as a plural acronym. So, in the line 192 it is necessary to change the verb "presents" to "present".

"...we concluded that Brc presents an increased tolerance to several antibiotics..."

Experimental design

The experimental design is solid and consistent with the purpose of the work. However, how Brc were obtained is not very clear. So, I would like to suggest I would suggest a more detailed statement about it.

Line 108 "Brc were collected by careful aspiration of the biofilm bulk fluid 24 hours after media replacement, as described above."

Validity of the findings

The provided data is robust, statistically valid and well controlled. However, there are some typos in the text and figures regarding the symbols used to demonstrate statistical difference.

In the legend of Figure 1. is missing to cite what means "***", and there is a "g" instead gama.

"...Brc are represented with * (* p <0.05; ** p <0.01), and between Brc and their planktonic
counterparts with γ (gγγ p <0.001)."

The same happens in the legend of Figure 2.

"...and between Brc and their planktonic counterparts with γ (g p <0.05; gγ p <0.01)."

·

Basic reporting

The article is well written and well illustrated in its figures.
Authors should carefully review bibliographic references, as is the case already in the first reference in which some authors' names are missing.

Experimental design

no comment'

Validity of the findings

no comment'

Additional comments

This article deals with a very important and interesting subject in microbiology, and in research on antimicrobial agents and bacterial resistance.
However the article is a simple article, and would fit well as brief communication. Their data is very well grounded and clear but although very important they indicate a lot to be initial data than would be a larger project.
It could also address a little more grounding in the introduction of respect for the importance of studying brc, as well as the importance of making an antimicrobial profile of the different types of S. epiderminidis cells

---

## Round 0.2 · accepted · Accept

I sincerely appreciate your time and effort in modifying your manuscript as per reviewers comments.

# ·

Basic reporting

The authors readily accepted the reviewers' suggestions and made the necessary changes to improve the manuscript. Therefore, I recommend the publication of this paper.

Experimental design

No comments.

Validity of the findings

No comments.